# Advanced Characterization of Hemp Flour (*Cannabis sativa* L.) from Dacia Secuieni and Zenit Varieties, Compared to Wheat Flour

**DOI:** 10.3390/plants10061237

**Published:** 2021-06-18

**Authors:** Iulian-Eugen Rusu, Romina Alina Marc (Vlaic), Crina Carmen Mureşan, Andruţa Elena Mureşan, Miuţa Rafila Filip, Bogdan-Mihai Onica, Kádár Balázs Csaba, Ersilia Alexa, Lidia Szanto, Sevastiţa Muste

**Affiliations:** 1Food Engineering Department, Faculty of Food Science and Technology, University of Agricultural Science and Veterinary Medicine Cluj-Napoca, 3–5 Calea Manaştur Street, 400372 Cluj-Napoca, Romania; iulian.rusu@usamvcluj.ro (I.-E.R.); andruta.muresan@usamvcluj.ro (A.E.M.); balazs-csaba.kadar@usamvcluj.ro (K.B.C.); lidia.szanto@usamvcluj.ro (L.S.); sevastita.muste@usamvcluj.ro (S.M.); 2Raluca Ripan Institute for Research in Chemistry, Babeş-Bolyai University, 30 Fântânele Street, 400294 Cluj-Napoca, Romania; miuta.filip@ubbcluj.ro; 3Department of Environmental and Plant Protection, University of Agricultural Science and Veterinary Medicine, 400372 Cluj-Napoca, Romania; bogdan.onica@usamvcluj.ro; 4Department of Food Control, Banat’s University of Agricultural Sciences and Veterinary Medicine “King Michael I of Romania”, 300641 Timisoara, Romania; alexa.ersilia@yahoo.ro

**Keywords:** hemp flour, proximate composition, micro and macro elements, fatty acids, carbohydrate

## Abstract

The advanced characterization of flour from hemp seeds (edible fruits of *Cannabis sativa* L.) from the Dacia Secuieni and Zenit varieties, compared to wheat flour, was studied in this research. The aim was to present the characterization of 2 varieties, out of the 70 accepted in Europe, for human consumption. The varieties selected from hemp meet the THC level requirement (0.3 or 0.2% of the dry weight of the reproductive part of the female flowering plant) in seeds. Hemp flour was obtained by grinding. The flour samples were evaluated for physicochemical parameters (moisture, crude protein, lipids, ash, crude fiber), the content of micro and macro elements, fatty acids, amino acids, and carbohydrates. The total proteins in hemp flour are found in larger quantities by over 35% compared to wheat flour, and the lipids reach the threshold of 28%. The amount of mineral substances exceeds 3% in hemp flour, and the fibers exceed 26%, compared to 0.61% for wheat flour. The predominant mineral substances were K, Ca, Mg, *p*, Fe, and Mn. The predominant fatty acids were the unsaturated ones, predominantly being linoleic acid, followed by oleic and gamma-linoleic acid. In the case of amino acids, the highest amount is found in glutamic acid for hemp flours. As for carbohydrates, sucrose is found in the largest amount, followed by glucose and fructose. In conclusion, hemp flours have superior non-traditional characteristics to wheat flour, being a potential raw material for fortifying food or using them as such, having beneficial effects of consumption on the proper functioning of the human body.

## 1. Introduction

*Cannabis sativa* L. belongs to the Cannabaceae family and is an herbaceous, anemophilous plant and is known as hemp, being one of the oldest cultivated plants. Most of the existing data show that they were first cultivated in Central Asia, and in Europe, the first available data on hemp are from the Bronze Age [1,2]. Hemp cultivation (*Cannabis sativa* L.) has the highest industrialization capacity of all technical plants, where everything is used, and nothing is thrown away [3]. According to Farinon et al. [4], hemp is cultivated for seed and fiber, being divided into two subspecies: *Cannabis sativa* ssp. Culta represents cultivated hemp and *Cannabis sativa* ssp. Spontanea represents wild hemp. At the present moment, the domestic form is cultivated both in Asian countries and in Europe, the United States, Canada, and Africa. It is cultivated for the diversity of fields of application such as agriculture, animal feed, pharmaceutical industry, food industry, and construction [3].

The main factor that causes confusion is related to the presence of the two phytochemicals specific to this plant, delta−9-tetrahydrocannabinol (THC) and non-psychoactive cannabidiol (CBD), the only psychoactive and toxic compound of the plant. These compounds belong to the class of cannabinoids and the family of terpenophenolic compounds. They are synthesized, then collected, and finally stored in specialized tiny secretory epidermal glands (glandular trichomes) [5,6]. These substances are absent in the seeds but are present in the inflorescence of the female plant, leaves, and stems [7,8,9]. The presence of these substances in the seeds could be given only by contamination [10].

From this point of view, following the awareness of the psychotropic activity of THC and the harmful effects of the consumption of hemp flowers and leaves, on human health, many countries have begun to take measures to stop consumption. The first measures were taken by the USA and Canada, and then in Europe. Starting with 2013, with the appearance of the EU regulation no. 1307/2013, the cultivation of *C. sativa* L. hemp was reintroduced for industrial purposes and only those plants with low THC levels (does not exceed 0.2% of the dry quantity, weight of leaves, and flowering parts) [7]. After 2013, the legislation was updated in 2016 in Italy (law no. 242/2016) [11] and in 2018 in the USA (farm bill) [12].

The total amount of lipids reported for hemp seeds is between 25% and 35% [13,14,15,16]. The predominant fatty acids (70–80%) in hemp seeds are the polyunsaturated ones (PUFA), being also present the saturated fatty acids (SFA), but in a smaller amount [17]. It should be noted that hemp seeds contain essential fatty acids [4], which respects the ideal ratio between n−6/n−3, varying between 3:1 and 5:1. This report is recommended for maintaining optimal health and for preventing cardiovascular disease, neurodegenerative diseases, or some cancers [14,16,18,19].

Proteins, reported in an amount ranging from 20–25%, are easy to digest and rich in essential amino acids [13,14,15,16,17,20,21]. The main proteins identified in hemp seeds are the storage protein albumin (a globular protein) and edestin (a legume) [22,23]. These proteins have a high digestibility [24], and the denaturation temperature is 92 °C [25]. According to the observations made by Lin et al. [26], a heat pretreatment at 100 °C for 15 or 30 min improves the digestibility of hemp proteins. From a nutritional point of view, the quality of hemp seed proteins is distinguished by the presence of essential amino acids, bioavailability, and digestibility [13,14,21,22,23,27].

Total carbohydrates are found in hemp seeds in a proportion of 20–30%, and a large part of them are dietary fiber, mainly insoluble [14,21]. However, there are authors who have observed that whole fibers in hemp seeds can negatively affect the digestibility of proteins [28,29,30]. It is important to remember that the consumption of dietary fiber brings many benefits to human health. These benefits include improved insulin sensitivity, are considered a functional product and have probiotic activity, may decrease appetite and high food intake, and reduce obesity and diabetes. Last but not least, dietary fiber reduces total blood cholesterol and low-density lipoproteins, and because they are not digested in the small intestine, they reach the large intestine, where they are fermented by the gut microbiota, producing short-chain fatty acids that have anti-inflammatory properties and anticancer [31].

The total mineral content is indicated by ash, which is the anorganic matter of a sample. In the case of hemp seeds, it is between 3.7% and 5.9%, depending on the variety, environmental conditions, compost, the use of various fertilizers in the soil, as well as the use of various fertilizers. These substances fall into the category of micronutrients because the food requirement is small (1–2500 mg/day depending on the type of mineral), but very important, playing an essential role in the proper functioning of the human body. The main macroelements reported in hemp seeds are phosphorus (*p*), magnesium (Mg), potassium (K), sodium (Na), and calcium (Ca), and in the category of trace elements, the majority were iron (Fe), zinc (Zn), manganese (Mn) and copper (Cu) [3,14,15,20].

With the awareness of the high nutritional value of hemp seeds, interest in the food industry began to grow. There is still confusion among the population about “industrial hemp” and “drug hemp”, even though there is legislation on this subject and sufficient sources of information. However, the negative reputation of the drug hemp influences consumption and investment for new perspectives [4].

Although there is significant information in the literature on the nutritional composition of hemp seeds in general, there are no specific studies on the varieties allowed for consumption and their functional potential. Currently, there are about 70 varieties of hemp *C. sativa* L., prerequisites and regulated in the European Union Plant Variety Database [4,32]. The largest hemp producers in the world are Canada, France, the Netherlands, Lithuania, and Romania [12].

Thus, this study aims to characterize two varieties of hemp out of the approximately 70 accepted in Europe, the Dacia Secuieni variety and the Zenit variety (THC < 0.2%), compared to wheat flour to illustrate the benefits of hemp flour and to open new directions of research, hemp can be used as an adjunct in the production of functional foods. One of these directions can be the bakery industry, being one of the basic branches of the food industry.

## 2. Results and Discussion

### 2.1. Results on Composite Flowers

#### 2.1.1. Physico-Chemical Analyses

Table 1 shows the physicochemical analyses performed on the wheat flour and hemp flour from the Dacia Secuieni variety and the Zenit variety. The wheat flour used has a higher humidity (9.2%) compared to the two types of hemp flour of the two varieties, within the maximum allowed limits (14.5%) according to order no. 328/23.09.2003 [33]. In the case of hemp flour, Vonapartis et al. [17] report moisture of hemp seeds between 1.1% and 7.2%, depending on the 10 varieties studied. The closest results to those reported by us are those of Lan et al. [20] between 4.1% and 4.3% for the 10 varieties of industrial hemp grown in North Dakota, USA. The humidity of the studied seeds can even reach up to 9.2% [13].

The total proteins from hemp flour are found in larger quantities by over 35% compared to wheat flour. The highest percentage is present in hemp flour of Dacia Secuieni variety, 25.19%. In the case of wheat flour, the percentage being only 13.04%. Values of 25.6% are also reported by Mattila et al. [21] or 24.8% for the hemp variety Finola [14] or Fedora [16]. Consequently, hemp seeds are considered a valuable and rich source of protein, having a similar or higher amount to other high protein products such as buckwheat seeds (27.8%), flax seeds (20.9%) [21], chia seeds (18.2–7.7%) [34], or quinoa (13.0%) [21].

Regarding the amount of lipids, there is a very high lipid content in the case of hemp flour 28.32% for Zenit variety, respectively 29.27% for Dacia Secuieni, compared to wheat 0.26%. Similar values were reported by Vonapartis et al. [17] and Mattila et al. [21]. However, there are studies that have reported lower lipid values of 24.5% [16] or higher than 35.9% [20].

The ash content in wheat flour is 0.55%, within the limits imposed by the classification of flour types according to the amount of ash [35]. Regarding the content of the two types of hemp flours, a significant increase is observed compared to that of wheat. The difference between the Dacia Secuieni and Zenit varieties is very small, respectively, 3.15% for the Dacia Secuieni variety and 3.10% for the Zenit variety. The lowest percentage of fiber is found in wheat flour (0.61%), its content increasing significantly for the two types of hemp flour. Values of 3.7% were also reported by Mattila et al. [21], but in most studies, the content of fiber was reported between 5% and 6% [14,15,16,17,20,21].

Following the results of the statistical analysis, it is observed that wheat flour has, except for moisture, a significantly lower value of crude protein compared to both varieties of hemp flour, between which there are no significant differences. The same scenario is repeated for lipids, crude fiber, and minerals, where wheat flour has the lowest values with significant differences from hemp flour variants.

Following the data in Table 2, we can see that there is a negative correlation between lipids and wheat flour. There is a negative correlation in the case of moisture, minerals, and crude fibers with wheat flour but also in the case of moisture with the two varieties of hemp flour. At the same time, we notice that there is a positive correlation in the case of the following variables: moisture and wheat flour, crude protein and hemp flour Dacia Secuieni, lipids and hemp flour of both varieties, mineral substances from hemp flour of varieties Dacia Secuieni and Zenit fiber and with Dacia Secuieni hemp flour, but also with Zenit variety.

#### 2.1.2. Content of Micro and Macro Elements

Following the analyses performed, 12 mineral elements were determined: copper (Cu), cadmium (Cd), chromium (Cr), nickel (Ni), lead (Pb), zinc (Zn), iron (Fe), manganese (Mn), calcium (Ca), magnesium (Mg), potassium (K), and phosphorus (*p*). The results obtained (Table 3) confirm that the highest amounts of minerals were reported in the case of K. The highest share is found in the case of hemp flour of the Zenit variety. In addition to K, a significantly higher content was identified for Ca, Mg, and *p*, also present in a higher share in hemp flour of the Zenit variety. The smallest amounts of minerals present in flours were given by Cd (only in the case of Zenit hemp flour), Cr, Ni, and Pb. Compared to the samples of hemp flour, the sample of wheat flour records the smallest amounts of mineral elements, some of which are even absent. These high values of mineral substances support the proposal to fortify wheat flour with hemp flour.

Following food requirements, mineral substances can be classified into macro elements in the case of minerals required in the amount of >50 mg/day, including K, *p*, Ca, Mg, and Na; and trace elements or trace elements in the case of minerals required in the amount of <50 mg/day including Cu, Fe, Zn, and Mg. Although the total amount of minerals has been studied by many authors, the micro and macroelement profile of hemp seeds is less studied. Most studies show that this profile is largely influenced by variety, environmental conditions, soil, and its mineral composition, and last but not least by the use of fertilizers and their type [3,14,15,16,20,21]. Comparing the results obtained by us with those in the literature, we can say that most of the macroelements found in hemp seeds were *p*, K, Mg, Ca, and Na, and in the case of microelements were Fe, Zn, and Cu. These values can be compared to those obtained for the Dacia Secuieni and Zenit varieties [3,14,15,20]. Following these results and the dietary reference values of EFSA [4], hemp seeds are considered a rich source of *p*, K, Ca, Mg, Fe, Cu, Zn, and Mn. Of note is the amount of *p*, which is found in the amount close to that of flax seeds (*Linum usitatissimum* L.), with average values of 461.35 mg/100 g, these being considered optimal sources of *p* [36]. The amount of K, the predominant mineral substance in hemp seeds, is found in much higher amounts than in flax seeds (568.91 mg/100 g) [36] or hazelnuts (863 mg/100 g), which are considered a source excellent from this macronutrient [37]. Even if the Na level is low, the high K/Na ratio is considered to have a cardioprotective effect. Mg, indispensable in cardiac functions, is found in amounts close to or higher than those reported in walnuts 381–443 mg/100 g, walnuts being on the list of foods recommended for high Mg content [4]. Among the microelements, Fe stands out due to its importance for human health and due to its widespread nutritional deficiencies [4]. The difference between the Dacia Secuieni and Zenit varieties is quite large 216.22, respectively 89.47 mg/100 g. These differences may be due primarily to the high level of iron in the soil, as observed by Mihoc et al. [3] for the varieties Zenit, Diana, Denise, Armanca, and Silvana. Lan et al. [20] sublimated low grain content compared to hemp seeds. This aspect can be noticed in our case as well, the wheat flour having only 5.77 mg/100 g Fe. In the same study, Lan et al. [20] stated after calculations on the daily reference intake for 30 g of hemp seeds that these are excellent sources of Fe, Mn, Zn, Cu, *p*, and Mg, these being found in larger quantities after peeling. Considering the toxicity of Cd on the human body and the studies on hemp seeds where the maximum allowed limit is far exceeded [3,21,38], we can observe that in the flours studied by us Cd, was not identified, except for those of the Dacia Secuinei variety, where the identified value is 0.002 mg/100 g.

Following the Pearson correlation (Table 4) regarding the significant values between micro and macroelements in the flour samples, we observe a very suitable correlation between wheat flour with the elements Cu, Cr, Pb, Zn, Fe, Mn, Mg, K, and *p*, between Dacia hemp flour Szeklers with Fe and Mn, and in the case of hemp flour of Zenit variety with the elements Cd, Cr, Ni, Ca, and Mg.

#### 2.1.3. Fatty Acids Content

Regarding the fatty acid content, nine saturated and unsaturated fatty acids were identified in wheat flour, hemp flour of Zenit, and Dacia Secuieni varieties, respectively: The saturated fatty acids identified are myristic acid, palmitic acid, stearic acid, margaric acid, arachidic acid. The identified unsaturated fatty acids are oleic acid, linoleic acid, gamma-linolenic acid, and palmitoleic acid.

According to the data in Table 5, the most significant percentages are given by the presence of linoleic acid in both wheat flour (43.99 g/100 g) and hemp flour of Zenit varieties (38.61 g/100 g) and Dacia Secuieni (38.83 g/100 g). The presence of palmitic acid (22.16 g/100 g) and oleic acid (17.36 g/100 g) also have a high percentage in wheat flour.

The low percentage fatty acids in wheat flour were: myristic acid, stearic acid, gamma-linoleic acid, margaric acid, palmitoleic acid, arachidic acid. In the case of the presence of acids in hemp flour, a significantly high percentage was given by oleic acid 23.6 g/100 g for the Dacia Secuineni variety and 20.58 g/100 g for the Zenit variety, respectively; and gamma-linolenic where their percentage being close. As in the case of wheat flour, myristic, margaric, palmitoleic, and arachidic acids are found in very small percentages.

The results in the literature are similar to those obtained by us. Thus, hemp seeds are characterized by a high content of unsaturated fatty acids and low amounts of saturated fatty acids. Moreover, unsaturated fatty acids are found in amounts of up to 90% [17]. They can be monounsaturated (MUFA) and polyunsaturated (PUFA). From the first category, the majority is oleic acid (18:1, n−9, OA), with values between 20.58% and 23.60% for hemp flours and 17.36% in the case of wheat flour. High OA values were reported for the Canadian Joey variety (18.78%) [20], but low values of up to 8.42% were also reported for the Italian Finola variety [28]. The main one (PUFA) is represented by linoleic acid (18:2, n−6, LA), with statistically insignificant values (38.83% and 38.61%, respectively) between the two studied varieties. LA is followed by γ-linoleic acid (18:3, n−6, GLA) in a proportion of 16.46% in the case of the Dacia Secuieni variety and 18.61 in the case of the Zenit variety. These two polyunsaturated oils belong to the category of essential fatty acids (EFA). They are essential because they cannot be synthesized and must be obtained from nutrition, as they are necessary for the proper functioning of the human body. Therefore, hemp seeds are considered oppressive sources of EFA [11,14,17,28]. As for saturated fatty acids (SFA), they do not exceed 23% of the total fatty acids. Palmitic acid (16:0, PA) was the main SFA with amounts between 11.92% and 13.02%, much lower than those reported for wheat flour (22.16%). In the SFA category, PA is followed by stearic acid (18:0, SA), with values between 5.98% and 6.23%, respectively, for hemp flour varieties. In the literature, PA was reported between 2% and 9%, and SA between 2% and 3.9%. These contradictory differences were justified by the type of variety [4,11,14,16,17,20,28,39]. Irakli et al. [11] in their study of six industrial hemp varieties in Greece, conducted for three consecutive years, showed major differences in AF, the contribution of genotype to variance being 99.6% for γ-linolenic acid, 91.2% for OA and 86.2% for PA, respectively.

Following the results of the Pearson correlation (Table 6), we notice a very suitable correlation in the case of wheat flour with the fatty acids contained but also in the case of Dacia Secuieni hemp flour regarding the content of myristic acid and oleic acid.

#### 2.1.4. Amino Acids Content

From a nutritional point of view, the quality of proteins is given by their composition in amino acids and their bioavailability. The amino acids identified in both hemp flour and wheat flour are: lysine, isoleucine, phenylalanine, these being part of the category of essential amino acids; as well as arginine, glutamic acid, cysteine, tyrosine, these being part of the category of non-essential amino acids; according to Table 7.

In the largest quantity in hemp seeds, glutamic acid is found both in the Dacia Secuieni variety and in the Zenit variety, according to Table 7. Instead, in wheat flour, the quantity is very small, 0.005 g/100 g. Glutamic acid is followed by arginine with values of more than 2.5 g/100 g of hemp flour and in the largest amount in wheat flour with values of 0.054 g/100 g. Values not to be neglected for hemp flour are also reported for lysine, phenylalanine, tyrosine, and in smaller amounts also for cysteine. In the case of wheat flour, the values for these amino acids are quite low. The results obtained are consistent with those reported by other authors, agreeing that hemp seeds contain all the essential amino acids (EAA) necessary for the proper functioning of the human body. At the same time, the predominant amino acid was glutamic acid with values between 3.74% and 4.58%, followed by arginine with values between 2.28% and 3.10% [13,14,15,21]. Given the quality of hemp seed proteins, several authors compared the amino acids in these seeds with the amino acids in soy and casein, which are considered important and high-quality sources of amino acids. From these reports, it was concluded that hemp seeds have a significant amount of sulfur-containing amino acids, even higher than in the case of casein or soy [14,22,23].

Following the results of the Pearson correlation (Table 8), we notice a very suitable correlation in the case of wheat flour with the fatty acids contained but also in the case of Dacia Secuieni hemp flour regarding the content of myristic acid and oleic acid.

#### 2.1.5. Carbohydrate Content

The amount of individual carbohydrates identified in hemp flours and wheat flour are sucrose, glucose, and fructose. The highest amount was identified in the case of sucrose, followed by glucose and fructose, regardless of the type of flour studied, according to Table 9. The amount of total sugars is found in a higher amount in hemp flours compared to wheat flour. In the case of Dacia Secuini hemp flour, a quantity of sucrose of over 18 mg/g can be noticed, and in the case of flour of the Zenit variety, it is found in quantities of 7 mg/g. In the case of glucose and fructose, the higher share is for Zenit flour. The amount of individual sugars is not studied in the literature. It is recalled that in most hemp seeds, most carbohydrates are represented by dietary fiber [4].

Authors such as Callaway et al. [14] studied the total carbohydrate content with results of 27.6 g/100 g for the Finola variety, and Mattila et al. [21] reports carbohydrate amounts for hemp seeds similar to those for flax seeds (34.4 g/100 g and 29.2 g/100 g, respectively).

In the case of wheat flour, the results obtained are similar to those reported by Schmidt et al. [40], who studied the total carbohydrate content of wheat flour and rye to know exactly these compounds, with an important role in feeding people with irritable bowel syndrome. In the case of the study by Manoharlal et al. [41] for Indian wheat flour, the results were lower than those reported by us. The results of the Pearson analysis (Table 10) show a positive correlation for the two varieties of hemp, in the case of fructose.

## 3. Materials and Methods

### 3.1. Materials

The study material used to perform the experiments consists of wheat flour sample, type 550, and hemp samples (*Cannabis Sativa* L.) from Dacia Secuieni and Zenit varieties, which were ground to obtain hemp flours.

Wheat flour was purchased from a local store. The husked seeds from which the hemp flour was obtained were purchased from the Secuieni Agricultural Research and Development Station, Neamț.

The Dacia Secuieni hemp variety was cultivated at S.C.D.A. Secuieni, being approved in 2011. It is characterized by stems with a length of 1.8–2.8 m in the culture for fiber and 3.5–4.5 m in the culture for seed. It achieves a high production of stems, up to 12 t/ha, with a fiber content of 30–31%. The seed production made by the Dacia Secuieni variety is 1000–1200 kg/ha, corresponding to a vegetation period of 145–160 days.

The Zenit hemp variety was approved in 2000. This variety has a vegetation period of 120–130 days in the case of seed cultivation and 90–110 days in the case of fiber cultivation. The seed production made by the Zenit variety is 1200 kg/ha, and the production of stems, 8–9 t/ha.

### 3.2. Proximate Composition

Proximate compositions were determined according to the standards in force: moisture was determined according to SR ISO 712:1999 [42]. Nitrogen (N) content was determined by the Kjeldahl apparatus, and crude protein was calculated using 5.7 as N conversion factor for vegetable products protein (SR ISO 1871/2002) [43]. The lipid content was determined according to SR ISO 6492:1999 [44], ash (mineral content) was determined according to Romanian official methods STAS 90/1988 [45], and the content of Kurdish fibers was determined according to ISO 5498:1981 [46].

### 3.3. Determination of Micro and Macroelements

Determination of the content of micro and macro elements by atomic absorption spectrometry was determined according to SR EN 14082:2003 [47].

### 3.4. Determination of Fatty Acid Composition

An application was made using the GC-MS system (gas-chromatograph coupled with a mass spectrophotometer) of a sensitive analytical method to be able to identify fatty acids from composite flour samples. With the help of the BF3-MeOH method, the methyl esterification of the samples used in the analysis was performed. This occurred after the alkaline hydrolysis process. For a 20 mg oil sample, 2 mL 0.5 mol/L methanol solution was added, and the mixture was heated for 7 min at 100 °C. After cooling, 3 mL of BF3-MeOH having a concentration of 14% was added, and the vessel was heated for 5 min at 100 °C. After cooling, 7 mL of saturated NaCl solution was added, after which the solution was extracted. The resulting hexane layer (2 mL) was used as a solution for gas chromatography [48]. The FAME analysis was performed based on GC-MS QP 2010 using the Shimadzu system. It was equipped with an automatic injector with and without splitting the mobile phase flow split/splitless. Separation was performed using a Zebron ZB-FFAP capillary column (60 m × 0.25 mm ID, 0.25 μm film thickness). Helium was used as a carrier gas with a flow rate of 1.99 mL/min with a split ratio of 1:10. The injector temperature reached was 250 °C. The oven temperature was programmed for 10 min at 140 °C, after which it was raised to 250 °C with a rate of 7 °C/minute, after which the final temperature was maintained for 10 min. The control of the GC—MS operation was possible thanks to the software from LabSolution. MS spectra were assimilated in a range having a width of 40–500 *m*/*z*, the ion source temperature was 210 °C, the interface temperature was 255 °C, the solvent was interrupted in 3 min, the scanning speed was 2500 units/s, and the actuation time was 0.2 s. The peaks of the fatty acid methyl esters were determined by comparing the equivalent chain length with the retention time. FAME standards from Supelco Inc., Bellefonte, PA (Supelco 37 component FAME Mix), and other reagents from Merck, Germany, were used. Three determinations were performed.

### 3.5. Determination of Amino Acids

Determination of amino acids using high-performance liquid chromatography (HPLC) involves their acid hydrolysis in the presence of 6 M HCL except for sulfur amino acids, identification, and chromatographic dosing where the DIONEX ICS—3000 amino acid analyzer is used. Take 0.5 g of the sample and hydrolyze with 10 mL of 6 M HCl at 100 °C for 24 h. The sample is filtered through a 0.2 μm Millipore filter, and the sample is diluted in a ratio of 1:10 or 1:1000 with 0.1 N HCl depending on the nature of the sample, finally injected into the chromatograph. Chromatographic conditions: chromatographic column pre-column AMINOPAC PA10 (2 × 50 mm, *p*/N 055407). Mobile phases: E1: water; E2: 250 mM NaOH and E3: 1 M NaAc, gradient, mobile phase flow (0.25 mL/min). Reference electrode: pH/Ag/AbCl, column temperature being 30 °C.

### 3.6. Determination of Individual Carbohydrate

The HPLC-RI method consists of separating at a temperature of 70 °C on a column of CARBOSep COREGel 87 C (300 × 7.8 mm) with a CARBOSep 87 C cartridge and CARBOSep COREGel 87 C cartridge. The mobile phase was represented by ultrapure Millipore water, and the flow rate was 0.5 mL/min., For the determination of glucose, sucrose, and fructose, and the injector volume was 20 µL. The process of extracting carbohydrates from the samples studied was performed in water. Weigh 3 g of the sample and grind it well, then add 8 mL of extraction solvent water. The mixture was ultrasonified for 15 min and then centrifuged for a period of 20 min at 4500 rpm. The supernatant was filtered through a 0.45 μm filter and injected into HPLC. Carbohydrate amounts were expressed in mg carbohydrate/100 g sample. The following materials were used in the determination of carbohydrates: acetonitrile (HPLC purity), it was purchased from Merck (Darmstadt, Germany), the standards of resveratrol, fructose, sucrose, and glucose were purchased from the United States, from Milwaukee, Aldrich, Millipore ultrapure water (18.2 MΩ cm^−1^). The equipment used consisted of: Jasco HPLC chromatograph (Japan) for analysis, it is equipped with an HPLC pump (Model PU−980), a column thermostat Model CO−2060 Plus, ternary gradient unit Model LG−980–02, UV-VIS detector (Model UV−975), injection valve equipped with a 20 µL test loop (Rheodyne) and detector with refractive index Model RI−2031 Plus. The samples were injected with a Hamilton Rheodyne syringe (50 mL). The HPLC system was controlled, and the experimental data were performed using ChromPass software. For the construction of the calibration curve, a standard mixture of carbohydrates with different dilutions (5 concentrations) was used, being between 333 and 2000 µg/mL. This HPLC-RI method developed by Filip et al. [49] was improved and optimized to identify and quantify the carbohydrates in flours samples.

### 3.7. Statistical Analysis

Statistical data analysis was performed using statistical interpretation programs R Studio version 3 [50] and StatSoft Statistics version 12 (STATSOFT, 2002). With the help of the StatSoft Statistics program version 12, Pearson correlations were made, and through the R Study program version 3, the Fisher LSD test was performed using the “agricolae” package [51].

Pearson correlations. The correlation in the broadest sense is defined as a measure of an association between variables. Changing the size of one variable in correlated data is associated with changing the size of another variable, either in the same direction (positive correlation) or in the opposite direction (negative correlation). The term correlation is used in the context of a linear relationship between two continuous variables and expressed as a Pearson product-moment correlation [52].

## 4. Conclusions

Given that in recent years the interest for consumption in the food industry of hemp seeds has increased, it is of great interest to know the chemical composition of the varieties allowed from a legislative point of view. The two varieties of hemp studied: Dacia Secuieni and Zenit meet the THC level requirement (0.3% or 0.2% of the dry weight of the reproductive part of the female plant in flowering). Analyzes showed that these varieties have an increased nutritional value, a high content of protein, fat, minerals, fiber, essential fatty acids, and essential amino acids. All these values, much lower than those found in wheat flour, make the hemp flour from this type of seeds much more qualitative. The entire composition of hemp seeds, but emphasizing the high amount of essential fatty acids and essential amino acids, it is recommended to use hemp seed flour for beneficial effects on the proper functioning of the human body. Hemp flour is an important vector in the food industry because it can be used in many different food technologies, such as beer, sugar, or the bakery industry.

## Figures and Tables

**Table 1 plants-10-01237-t001:** Physicochemical properties of wheat flour and hemp flour.

Samples	Moisture (%)	Crude Protein (%)	Lipids (%)	Ash (%)	Crude Fiber (%)
Wheat Flour 550	9.2 ± 0.70 ^a^	13.04 ± 0.04 ^b^	0.26 ± 0.04 ^b^	0.55 ± 0.08 ^b^	0.61 ± 0.03 ^b^
Hemp flour variety D.S.	4.49 ± 0.20 ^b^	25.19 ± 0.52 ^a^	29.27 ± 0.05 ^a^	3.10 ± 0.04 ^a^	26.10 ± 0.04 ^a^
Hemp flour variety Z.	4.98 ± 0.032 ^b^	22.38 ± 0.59 ^a^	28.32 ± 0.16 ^a^	3.15 ± 0.05 ^a^	29.14 ± 0.04 ^a^

Identical superscripts letters within rows indicate no significant difference (*p* > 0.05); D.S.—Dacia Secuieni; Z.—Zenit.

**Table 2 plants-10-01237-t002:** Pearson correlation for physicochemical properties of flour sample.

Variable	Wheat Flour	Hemp Flour Variety D.S.	Hemp Flour Variety Z.
Moisture	0.996	−0.580	−0.416
Crude protein	−0.975	0.679	0.296
Lipids	−1.000	0.525	0.475
Minerals	−0.999	0.485	0.514
Crude Fiber	−0.995	0.414	0.582

D.S.—Dacia Secuieni; Z.—Zenit.

**Table 3 plants-10-01237-t003:** The content of micro and macroelements of wheat flour and hemp flour.

Samples (mg/100 g)	Cu	Cd	Cr	Ni	Pb	Zn	Fe	Mn	Ca	Mg	K	*p*
Wheat Flour	1.10 ± 0.040 ^b^	0 ^b^	0 ^c^	0 ^c^	0.14 ± 0.03 ^a^	4.60 ± 0.04 ^b^	5.77 ± 0.02 ^b^	4.69 ± 0.02 ^b^	463.00 ± 2.52 ^b^	131.33 ± 0.03 ^c^	1446.67 ± 0.06 ^b^	187.00 ± 2.00 ^c^
Hemp flour variety D.S.	9.98 ± 0.03 ^a^	0 ^b^	0.41 ± 0.03 ^b^	1.002 ± 0.02 ^b^	0.06 ± 0.02 ^b^	40.05 ± 0.01 ^a^	216.23 ± 0.04 ^a^	167.99 ± 0.77 ^a^	500.00 ± 2.52 ^b^	319.00 ± 2.52 ^b^	4052.00 ± 4.04 ^a^	350.00 ± 2.52 ^b^
Hemp flour variety Z.	10.31 ± 0.06 ^a^	0.002 ± 0.001 ^a^	1.24 ± 0.03 ^a^	3.84 ± 0.03 ^a^	0.06 ± 0.02 ^b^	38.88 ± 0.06 ^a^	89.47 ± 0.03 ^ab^	112.30 ± 0.89 ^ab^	715.33 ± 0.68 ^a^	599.67 ± 0.07 ^a^	4786.67 ± 0.08 ^a^	415.33 ± 0.02 ^a^

Identical superscripts letters within rows indicate no significant difference (*p* > 0.05); D.S.—Dacia Secuieni; Z.—Zenit.

**Table 4 plants-10-01237-t004:** Pearson correlation for micro and macroelements of flour sample.

Variable	Wheat Flour	Hemp Flour Variety D.S.	Hemp Flour Variety Z.
Cu	−1.000	0.473	0.527
Cd	−0.500	−0.500	1.000
Cr	−0.756	−0.189	0.945
Ni	−0.702	−0.266	0.968
Pb	1.000	−0.509	−0.491
Zn	−1.000	0.525	0.475
Fe	−0.801	0.919	−0.117
Mn	−0.942	0.762	0.181
Ca	−0.613	−0.378	0.991
Mg	−0.803	−0.114	0.917
K	−0.978	0.308	0.670
*p*	−0.961	0.240	0.721

D.S.—Dacia Secuieni; Z.—Zenit.

**Table 5 plants-10-01237-t005:** The content of fatty acids of wheat flour and hemp flour.

Samples [% of Total Fatty Acids]	Wheat Flour	Hemp Flour Variety D.S.	Hemp Flour Variety Z.
Myristic acid	1.61 ± 0.04 ^a^	0.24 ± 0.03 ^b^	0.92 ± 0.04 ^ab^
Palmitic acid	22.16 ± 0.04 ^a^	11.92 ± 0.05 ^b^	13.02 ± 0.06 ^b^
Stearic acid	3.96 ± 0.03 ^b^	5.98 ± 0.03 ^a^	6.23 ± 0.02 ^a^
Oleic acid	17.36 ± 0.03 ^b^	23.6 ± 0.03 ^a^	20.58 ± 0.03 ^ab^
Linoleic acid	43.99 ± 0.46 ^a^	38.83 ± 0.04 ^b^	38.61 ± 0.02 ^b^
Gamma—linolenic acid	5.41 ± 0.03 ^b^	16.46 ± 0.03 ^a^	18.61 ± 0.03 ^a^
Margaric acid	0.88 ± 0.03 ^a^	0.25 ± 0.03 ^b^	0.29 ± 0.03 ^b^
Palmitoleic acid	0.5 ± 0.03 ^a^	0.06 ± 0.03 ^b^	0.11 ± 0.03 ^b^
Arachidic acid	0.22 ± 0.03 ^b^	1.84 ± 0.03 ^a^	4,48 ± 0.04 ^b^
∑SFA	5.76 ± 0.03 ^a^	4.04 ± 0.03 ^b^	4,48 ± 0.04 ^b^
∑MUFA	8.93 ± 0.03 ^b^	11.83 ± 0.03 ^a^	10.35 ± 0.03 ^ab^
∑PUFA	24.69 ± 0.24 ^b^	27.64 ± 0.04 ^a^	28.61 ± 0.03 ^a^

Identical superscripts letters within rows indicate no significant difference (*p* > 0.05); D.S.—Dacia Secuieni; Z.—Zenit.

**Table 6 plants-10-01237-t006:** Pearson correlation for fatty acids of flour sample.

Variable	Wheat Flour	Hemp Flour Variety D.S.	Hemp Flour Variety Z.
Myristic acid	0.868	−0.864	−0.003
Palmitic acid	0.995	−0.582	−0.413
Stearic acid	−0.995	0.410	0.585
Oleic acid	−0.875	0.857	0.018
Linoleic acid	0.999	−0.469	−0.530
Gamma-linolenic acid	−0.988	0.362	0.626
Margaric acid	0.999	−0.539	−0.460
Palmitoleic acid	0.993	−0.602	−0.391
Arachidic acid	−0.999	0.453	0.546
ΣSFA	0.969	−0.695	−0.273
ΣMUFA	−0.860	0.871	−0.011
ΣPUFA	−0.969	0.279	0.689

D.S.—Dacia Secuieni; Z.—Zenit.

**Table 7 plants-10-01237-t007:** The content of amino acids of wheat flour and hemp flour.

Samples (g/100 g)	Wheat Flour	Hemp Flour Variety D.S.	Hemp Flour Variety Z.
Arginine	0.054 ± 0.002 ^b^	2.72 ± 0.096 ^a^	2.66 ± 0.075 ^a^
Lysine	0.005 ± 0.002 ^b^	1.04 ± 0.070 ^a^	0.99 ± 0.075 ^a^
Isoleucine	0.001 ± 0.0002 ^b^	0.82 ± 0.040 ^a^	0.79 ± 0.045 ^a^
Phenylalanine	0.003 ± 0.002 ^b^	1.04 ± 0.025 ^a^	1.03 ± 0.015 ^a^
Glutamic acid	0.005 ± 0.002 ^b^	4.13 ± 0.215 ^a^	3.98 ± 0.115 ^a^
Cysteine	0.011 ± 0.002 ^b^	0.37 ± 0.036 ^a^	0.34 ± 0.015 ^a^
Tyrosine	0.003 ± 0.002 ^b^	0.76 ± 0.045 ^a^	0.74 ± 0.045 ^a^

Identical superscripts letters within rows indicate no significant difference (*p* > 0.05); D.S.—Dacia Secuieni; Z.—Zenit.

**Table 8 plants-10-01237-t008:** Pearson correlation for amino acids of flour sample.

Variable	Wheat Flour	Hemp Flour Variety D.S.	Hemp Flour Variety Z.
Arginine	–0.99869	0.518265	0.480426
Lysine	–0.9937	0.538668	0.455029
Isoleucine	–0.9964	0.532294	0.464107
Phenylalanine	–0.99949	0.511854	0.48764
Glutamic acid	–0.99772	0.525348	0.47237
Cysteine	–0.99004	0.567307	0.422733
Tyrosine	–0.99599	0.51799	0.478

D.S.—Dacia Secuieni; Z.—Zenit.

**Table 9 plants-10-01237-t009:** The content of carbohydrates of wheat flour and hemp flour.

Samples (g/100 g)	Sucrose	Glucose	Fructose
Wheat Flour	5.54 ± 0.02 ^b^	2.03 ± 0.02 ^c^	1.03 ± 0.02 ^c^
Hemp flour variety D.S.	18.77 ± 0.03 ^a^	4.41 ± 0.02 ^b^	4.38 ± 0.03 ^b^
Hemp flour variety Z.	7.39 ± 0.03 ^ab^	7.10 ± 0.03 ^a^	6.38 ± 0.04 ^a^

Identical superscripts letters within rows indicate no significant difference (*p* > 0.05); D.S.—Dacia Secuieni; Z.—Zenit.

**Table 10 plants-10-01237-t010:** Pearson correlation for carbohydrate of flour sample.

Variable	Wheat Flour	Hemp Flour Variety D.S.	Hemp Flour Variety Z.
Sucrose	−0.60782	0.991592	−0.38377
Glucose	−0.84763	−0.03563	0.88326
Fructose	−0.92884	0.14372	0.785124

D.S.—Dacia Secuieni; Z.—Zenit.

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
