# Peer review of "Advanced Characterization of Hemp Flour (Cannabis sativa L.) from Dacia Secuieni and Zenit Varieties, Compared to Wheat Flour"

_plants, 2021, doi:10.3390/plants10061237_

Round 1
Reviewer 1 Report
The paper is on the composition of hemp seeds in comparison with wheat flour. a few major issues for you to consider:
There are already studies on the chemical composition of the hemp seeds. What are the advantages of your studies? Are they nessesary?
Why u did Pearson analysis? It seems to be inappropriate in such cases.
Data in the tables are strange. No units for them.
Overall, the paper is not well written. Too many corrections are needed.
The conclusions part is meaningless without much useful information.
Author Response
Comment: The paper is on the composition of hemp seeds in comparison with wheat flour. a few major issues for you to consider:
There are already studies on the chemical composition of the hemp seeds. What are the advantages of your studies? Are they nessesary?
Authors response: Thank you for encouraging comments.
In the literature there are some studies on several varieties of hemp seeds, or on hemp seeds in general, but not complex. In Europe there are 70 varieties accepted from a legislative point of view, but there are no studies on their advanced characterization. In the literature there is no such complex study on the characterization of any kind of hemp regarding the characteristics of the seeds. Hemp seeds are a potential ingredient for food fortification and advanced knowledge of chemical composition is required. Hemp seeds were compared with wheat flour to highlight their properties, compared to one of the most widely used raw materials in the world.
Comment: Why u did Pearson analysis? It seems to be inappropriate in such cases.
Authors response: Pearson correlation (linear correlation coefficient) measures the degree of connection between variables. This correlation was made to highlight and express in as much detail as possible the strengths and weaknesses of the properties of hemp seeds compared to wheat flour.
Comment: Data in the tables are strange. No units for them.
Authors response: Corrected
Comment: Overall, the paper is not well written. Too many corrections are needed.
Authors response: Improvements have been made. If they are not enough, please be more explicit in what needs to be changed.
Comment: The conclusions part is meaningless without much useful information.
Authors response: Corrected
Thank you once again for your valuable comments. I am available if there are any further queries.
--
Best regards,
Marc (Vlaic) Romina Alina et al.

Reviewer 2 Report
This is a well written paper. The English is good and the data are presented in a clear way. Thus one can quickly read the paper and understand it.
The experiments presented have been done in a good way, and the data appear to be reliable
The manuscript present new and useful data regarding the composition of hemp seeds. This is a useful addition to the literature. My view is the paper is suitable for publication in Plants after minor revision as outlined below
ln 42 claim to come from . This does not make sense change to something like '..existing data indicate it was first cultivated in....'
Author Response
Comment: This is a well written paper. The English is good and the data are presented in a clear way. Thus one can quickly read the paper and understand it.
The experiments presented have been done in a good way, and the data appear to be reliable
The manuscript present new and useful data regarding the composition of hemp seeds. This is a useful addition to the literature. My view is the paper is suitable for publication in Plants after minor revision as outlined below
Authors response: Thank you for encouraging comments.
Comment: ln 42 claim to come from . This does not make sense change to something like '..existing data indicate it was first cultivated in....'
Authors response: Corrected
Thank you once again for your valuable comments. I am available if there are any further queries.
--
Best regards,
Marc (Vlaic) Romina Alina et al.

Reviewer 3 Report
The manuscript can be published in Plants, but I have some concerns, and my comments are as below:
Abstract and introduction:
- The abstract must include data regarding the critical finds by the authors.
- The introduction must have a clear hypothesis and significantly develop the second paragraph of this manuscript.
- The last lines should melt down to the importance of this study.
Material and Methods
The methods are not properly defined and left open without proper details.
Results:
Are carelessly written and needs a proper sentence structure.
Discussion should include more information and references related to the relevant and related works.
Author Response
Comment: The manuscript can be published in Plants, but I have some concerns, and my comments are as below:
Authors response: Thank you for encouraging comments.
Comment: Abstract and introduction:
- The abstract must include data regarding the critical finds by the authors.
- The introduction must have a clear hypothesis and significantly develop the second paragraph of this manuscript.
- The last lines should melt down to the importance of this study.
Authors response: The abstract respects the requirements of the journal and provides a pertinent overview of the work. The purpose of the study is briefly presented, the result is brief and a short conclusion.
The explanations for the second paragraph continue in the third paragraph. No further details are presented as it is not the purpose of our study.
The last paragraph of the introduction presents the importance of the study.
Comment: Material and Methods
The methods are not properly defined and left open without proper details.
Authors response: The briefly described methods refer to standards or other manuscripts, where they are described in detail. This brief description was made to avoid plagiarism. The methods are standardized or a protocol has been followed which has already been described in detail in the manuscripts to which reference is made.
Comment: Results: Are carelessly written and needs a proper sentence structure.
Discussion should include more information and references related to the relevant and related works.
Authors response: Some improvements have been made. In the opinion of the authors, the results are well structured, arranged and commented in detail with reference to the relevant works existing in the specialized literature. Please give us more details on what we should improve.
Thank you once again for your valuable comments. I am available if there are any further queries.
--
Best regards,
Marc (Vlaic) Romina Alina et al.

Round 2
Reviewer 1 Report
only minor changes were done to the paper.
Reviewer 3 Report
The authors have incorporated the changes I mentioned; therefore, the manuscript can be accepted for publication after a careful spell check and correcting minor English language mistakes.